# Identification of Biomarkers Related to Metabolically Unhealthy Obesity in Korean Obese Adolescents: A Cross-Sectional Study

**DOI:** 10.3390/children10020322

**Published:** 2023-02-08

**Authors:** Sarang Jeong, Han-Byul Jang, Hyo-Jin Kim, Hye-Ja Lee

**Affiliations:** Division of Endocrine and Kidney Disease Research, Department of Chronic Disease Convergence Research, Korea National Institute of Health, Korea Disease Control and Prevention Agency, 187 Osongsaengmyeong 2-ro, Osong-eup, Cheongju-si 28159, Republic of Korea

**Keywords:** pediatrics, obesity, metabolically unhealthy obesity, MUO, metabolically healthy obesity, MHO, triglyceride-glucose index, TyG index, metabolic syndrome, MetS, diabetes mellitus, DM

## Abstract

Background: The current study aimed to screen for relationships and different potential metabolic biomarkers involved between metabolically healthy obesity (MHO) and metabolically unhealthy obesity (MUO) in adolescents. Methods: The study included 148 obese adolescents aged between 14 and 16. The study participants were divided into MUO and MHO groups based on the age-specific adolescent metabolic syndrome (MetS) criteria of the International Diabetes Federation. The current study was conducted to investigate the clinical and metabolic differences between the MHO and MUO groups. Multivariate analyses were conducted to investigate the metabolites as independent predictors for the odds ratio and the presence of the MetS. Results: There were significant differences in the three acylcarnitines, five amino acids, glutamine/glutamate ratio, three biogenic amines, two glycerophospholipids, and the triglyceride-glucose index between the MUO group and those in the MHO group. Moreover, several metabolites were associated with the prevalence of MUO. Additionally, several metabolites were inversely correlated with MHO in the MUO group. Conclusions: In this study, the biomarkers found in this study have the potential to reflect the clinical outcomes of the MUO group. These biomarkers will lead to a better understanding of MetS in obese adolescents.

## 1. Introduction

Despite growing health concerns, the prevalence of obesity in adults and children remains high worldwide [1]. Obesity is defined as excessive body fat accumulation [2]. Obesity is associated with cardiovascular disease (CVD) [3] and metabolic syndrome (MetS—diabetes mellitus (DM), hypertension (HTN), and dyslipidemia). In addition, MetS is associated with obesity [4] and insulin resistance (IR) [5]. However, obesity and MetS have different clinical features [6]. Therefore, it is important to understand and control obesity and MetS.

The prevalence of obese adolescents has an increasing trend [7]. According to the 2013–2017 obesity fact sheet of the Korea Health Promotion Institute, the prevalence of Korean obese adolescents increased from 11.6% in 2007 to 17.3% in 2017 [8]. Furthermore, obese adolescents are at a higher risk of MetS and CVD than non-obese adolescents [9]. Clinical factors are very important factors related to obesity and atherosclerosis [10]. Several previous studies have suggested that metabolic biomarkers are related to obesity [11], MetS [12], and CVD in adolescents [13]. However, periodic studies in adolescents are required because adolescence is a growing period that is sensitive to environmental and temporal changes [14]. Thus, more detailed research about sensitive biomarkers and regulatory pathways concerning obesity and MetS in adolescents is required. There are a few studies about biomarkers or regulatory pathways that are atherogenic or related to obesity. However, not all obese populations develop MetS [6]. In general, height and weight measurements are considered in obesity diagnosis [2]. Therefore, the diagnosis of MetS in the obese population requires consideration of other factors such as body fat composition, laboratory measurements, and metabolites. In this regard, obesity is classified as metabolically unhealthy obesity (MUO) and metabolically healthy obesity (MHO) [15].

Metabolites are crucial biomarkers that can be potentially used to observe dynamic physiological conditions corresponding to the clinical outcome of the patient [16]. Therefore, it is material to identify and/or quantify tools in adolescents to investigate the difference between MHO and MUO of biomarkers and regulatory pathways in obese adolescents.

The current study aimed to screen for relationships and different potential metabolic biomarkers involved between MHO and MUO in adolescents. We analyzed the difference between the two groups after adjusting for factors, such as age, sex, and body mass index (BMI) to exclude the effect of severe change by growth phase. Thus, the current study was conducted to investigate the clinical and metabolic differences between the adolescents in the MHO group and the adolescents in the MUO group.

## 2. Materials and Methods

### 2.1. Study Participants

The data were obtained from the Korean Children-Adolescents Study (KoCAS), conducted by the Korean National Institute of Health. The KoCAS subjects in this study were adolescents aged 14–16 years from Seoul and Gyeonggi provinces for whom clinical biomarker data were collected in 2012. Obesity was defined as a body mass index (BMI) >25 kg/m^2^ or being in the 95th percentile for age and sex according to the 2017 Korean growth standard [2] for children and adolescents. Among the adolescents, 143 who were severely obese adolescents aged 14–16 years with BMIs ≥ 99th percentile (or ≥30 kg/m^2^) were used in present study. Adolescents who participated in this study participated only with parental informed consent. The overall objective of this cross-sectional study was to identify early risk factors for obesity and associated metabolic disease in Korean adolescents. The current study excluded participants who did not have the data on confounding variables, such as nutrition intakes, biochemicals, and metabolites. The study participants were divided into MUO and MHO groups based on the age-specific adolescent MetS criteria of the International Diabetes Federation (IDF) [17]. The Institutional Review Board (IRB) of Korea National Institute of Health (KNIH; IRB No. 2020-07-05-P-A) approved the study protocol which complied with the Declaration of Helsinki.

### 2.2. Anthropometric Parameters

The anthropometric parameters of the study participants were measured in absence of clothing and shoes in the morning after overnight fasting for 12 h. The body weight and body fat percentage (BF%) were measured using a body composition analyzer (BC418; Tanita, Tokyo, Japan). The height was measured to the nearest 0.1 cm with a wall-mounted stadiometer (DS-102; Jenix, Seoul, Korea). The BMI was calculated as weight in kilograms divided by height in meters squared (kg/m^2^). The BMI z-score was calculated by BMI-for-age percentiles together with the Lambda-Mu-Sigma method of Cole and Green, which provides a way of obtaining normalized growth percentile standards based on The 2017 Korean National Growth Charts for Children and Adolescents [2]. The waist circumference (WC) was measured, with participants standing straight, using a flexible tapeline at the midpoint of the lower rib and the iliac crest to the nearest 0.1 cm. The hip circumference (HC) was measured using a flexible tapeline at the horizontal circumference of the highest point of the buttocks to the nearest 0.1 cm. The waist–hip ratio was calculated based on recorded WC and HC measurements. The systolic blood pressure (SBP) and diastolic blood pressure (DBP) were measured twice on the arm of participants in a seated position following a rest of at least 5 min using an automatic sphygmomanometer (HEM-907, OMRON Healthcare Co., Kyoto, Japan); the two measurements were then averaged. The basal metabolic rate (BMR; Kcal) was calculated using the equation of the Harris–Benedict formula: BMR = [66.47 (for males) or 65.51 (for females)] + [9.6 × weight (kg)] + [1.8 × height (cm)] − [4.7 × age] [18].

### 2.3. Biochemical Analysis

The blood samples were collected after an overnight fast of at least 12 h. The antecubital vein blood was collected in a vacutainer tube. The blood samples were centrifuged to obtain the plasma and serum samples, then stored at −80 °C. The levels of triglycerides (TGs), total cholesterol (T-cholesterol), high-density lipoprotein cholesterol (HDL-cholesterol), alanine aminotransferase (ALT), aspartate aminotransferase (AST), and glucose were measured using an autoanalyzer (model 7600Ⅱ; Hitachi, Tokyo, Japan). The low-density lipoprotein cholesterol (LDL-cholesterol) levels calculated using the equation of the Friedewald formula: LDL-cholesterol = total cholesterol—[HDL-cholesterol + (TG/5)].

### 2.4. MetS Diagnosis

In the present study, MetS was diagnosed using adolescent criteria of IDF [17]; the definition ages were 10–16 years. MetS was defined as central obesity (defined as WC but can assumed if BMI >30 kg/m^2^) plus at least 2 out of 4 criteria (No. 2–5). The MetS diagnostic criteria are as follows:WC ≥ 90th percentile or adult cut-off if lowerSBP of 130 mmHg or DBP of 85 mmHg or treatment with anti-hypertensive medicationTG ≥ 150 mg/dLHDL-cholesterol < 40 mg/dLFasting plasma glucose ≥ 100 mg/dL or known type 2 (T2) DM

### 2.5. Assessment of Nutrition Intake

In this study, participants were interviewed to obtain information about their nutrient intake. The dietary intake was assessed using a 3-day food record method. The data of food records were validated through their parents or caregiver. The study participants were asked to record their intake of meals and snacks, including beverages during a nonconsecutive period of 3 days (1 day of the weekend and 2 days of the weekdays). Furthermore, ingredients of meals, as well as portion sizes, were also recorded. A well-trained dietician reviewed and confirmed the written 3-day food record using food models during face-to-face interviews to increase precision in reporting. The research-based typical food intake data were used to calculate averages of energy, macronutrients, and micronutrients were analyzed through a food consumption survey database of the Korean National Health and Nutrition Examination Survey.

### 2.6. Metabolite Measurements

A total of 186 metabolites in the plasma of 148 participants were measured using AbsoluteIDQ^™^ p180 kit (Biocrates Life Sciences AG, Innsbruck, Austria). The data quality of each metabolite was checked based on the following criteria: (1) half of the analyzed metabolite concentrations in the reference standards > limit of detection and (2) half of the analyzed metabolite concentrations in the experimental samples. We excluded 32 metabolites that failed the quality criteria. Finally, a total of 154 metabolites (37 acylcarnitines (ACs), 20 amino acids (AAs), 8 biogenic amines (BAs), 81 glycerophospholipids (GPLs), and 8 sphingolipids (SPLs)) were analyzed with the AbsoluteIDQ™ p180 kit using the protocol described in the AbsoluteIDQ™ p180 user manual. The ACs, GPLs, and SPLs were quantified by flow injection analysis by tandem mass spectrometry using an ABI 4000 Q-Trap mass spectrometer (Applied Biosystems/MDS Sciex, Foster city, CA). The AAs and BAs were quantified by stable isotope dilution in a liquid chromatography-tandem mass spectrometry. The biocrates MetIQ software was used to control the entire assay workflow, from sample registration to the automated calculation of metabolite concentrations to the export of data into other data analysis programs. The metabolite concentration measurements in μmol/L (μM) units were automatically carried out with the MetVal™ software package (Biocrates Life Sciences AG, Innsbruck, Austria).

### 2.7. The Calculation of Homeostasis Model Assessment of Insulin Resistance (HOMA-IR) Index and Triglyceride-Glucose (TyG) Index

The HOMA-IR was calculated as the fasting glucose (mg/dL) × fasting insulin (μIU/mL)/405 [19]. The TyG index was calculated as the Ln [fasting glucose (mg/dL) × fasting TG (mg/dL)/2] [20].

### 2.8. Data Processing and Statistical Analysis

The statistical analyses were conducted using the Statistical Analysis System software version 9.4 (SAS Institute, Cary, NC, USA). The skewed variables were transformed logarithmically. For the comparison of categorical variables, a chi-squared test was conducted. The general linear model with a Bonferroni correction adjusting for confounding factors was used to compare the parameters collected from each group. Multivariate logistic regressions were conducted to investigate the metabolites as independent predictors for the odds ratio (OR) and the presence of the MetS in adolescents. The results are expressed as the means ± standard errors (SE). A two-tailed *p*-value of <0.05 was considered to be statistically significant. The Pearson’s correlation coefficients were calculated to measure the extent of correlation between pairs of variables.

## 3. Results

### 3.1. Participants

A total of 148 obese adolescents between the ages of 14–16 years and both sexes (male; *n* = 82 and female; *n* = 66) participated in the current study. Among the total participants, 74 adolescents belonged to the MUO group, while the other 74 adolescents belonged to the MHO group.

### 3.2. Anthropometric Parameters, Characteristics, and Laboratory Measurements of MUO and MHO in Adolescents

The anthropometric parameters, characteristics, and laboratory measurements of MUO and MHO in adolescents are shown in Table 1. Sex was not significantly different between the MUO and MHO groups. Weight, BMI, WC, HC, and body fat in the MUO group was higher than in the MHO group. However, the difference was not significant. The mean of height, SBP, DBP, fat-free mass (FFM), BMI percentage (BMI%), weight per age percentile (weight^th^), and BMR in the MUO group was significantly higher than in the MHO group (*p* = 0.0148, *p* < 0.0001, *p* < 0.0001, *p* = 0.0169, *p* = 0.0164, *p* = 0.0073, and *p* = 0.0480, respectively). The mean value of AST, ALT, red blood cell (RBC), hemoglobin (Hb), hematocrit (Hct), platelet (Plt), and insulin was not significantly different between the two groups. The white blood cell count and glucose levels in the MUO group were significantly higher than in the MHO group; however, these levels were in a normal range for adolescents. We adjusted for factors such as age, sex, and BMI variable, except for self-correction.

### 3.3. Plasma Lipid Profiles of MUO and MHO in Adolescents

The TGs in the adolescents in the MUO group were higher than those in the MHO group (*p* < 0.0001, Table 1). On the contrary, there was no significant difference in the T-cholesterol, LDL-cholesterol, and HDL-cholesterol levels between adolescents in the MUO group and those in the MHO group. We adjusted for factors such as age, sex, and BMI variable.

### 3.4. IR Assessment Index of MUO and MHO in Adolescents

The TyG index of the IR assessment index was significantly higher in the MUO group than in the MHO group (*p* < 0.0001, Table 1). However, the HOMA-IR was not significantly different between the two groups. We adjusted for factors such as age, sex, and BMI variable.

### 3.5. The Metabolites of MUO and MHO in Adolescents

When individual metabolites were considered, three ACs, five AAs, glutamine/glutamate (Gln/Glu) ratio, three BAs, and two GPLs were observed, which were significantly associated with a higher incidence of MUO in adolescents (Table 2). The acetylcarnitine (C2), hydroxy propionyl carnitine (C3-OH), and methyl glutaryl carnitine (C5-M-DC) of ACs were significantly different between the adolescents of the MUO group and those of the MHO group (*p* = 0.0184, *p* < 0.0001, and *p* = 0.0273, respectively). AAs such as alanine (Ala), glutamine (Gln), histidine (His), lysine (Lys), and serine (Ser) were significantly different between the adolescents of the MUO group and those of the MHO group (*p* = 0.0125, *p =* 0.0041, *p* = 0.0322, *p* = 0.0371, and *p =* 0.0061, respectively). The Gln/Glu ratio in the MUO group was significantly higher than in the MHO group (*p* = 0.0179). The BAs, such as kynurenine, methionine-sulfoxide (Met-SO), and spermidine, were significantly different between the adolescents of the MUO group and the adolescents of the MHO group (*p* = 0.0003, *p* = 0.0194, and *p* = 0.0346, respectively). The phosphatidylcholine (PC) diacyl (aa) (C32:2 and C34:1) of GPLs were significantly different between the adolescents of the MUO group and the adolescents of the MHO group (*p* = 0.0397 and *p* = 0.0402, respectively). The SPLs were not significantly different between the adolescents of the two groups.

### 3.6. Predictors of the MUO Adolescents’ Prevalence ORs on Significantly Different Metabolites

Predictors of the MUO adolescents’ prevalence ORs on significantly different metabolites are shown in Table 3. ORs for MUO prevalence were each fold higher in the upper quartiles of the C2 [OR; 1.606; 95% confidence interval (CI): 1.191–2.165; *p* = 0.0014], Ala [OR; 1.621; 95% CI: 1.194–2.200; *p* = 0.0014], Glu [OR; 1.364; 95% CI: 1.024–1.818; *p* = 0.0316], Gln/Glu ratio [OR; 0.735; 95% CI: 0.547–0.988; *p* = 0.0388], kynurenine [OR; 1.661; 95% CI: 1.222–2.259; *p* = 0.0008], Met-SO [OR; 1.983; 95% CI: 1.371–2.867; *p* = 0.0001], C34:1 [OR; 1.424; 95% CI: 1.057–1.917; *p* = 0.0180], HOMA-IR [OR; 1.360; 95% CI: 1.012–1.827; *p* = 0.0388], and TyG index [OR; 2.046; 95% CI: 1.476–2.835; *p* < 0.0001]. On the other hand, every other metabolite, except for these metabolites, showed no significant difference between the adolescents in the MUO group and the adolescents in the MHO group.

### 3.7. The Nutrition Intakes of MUO and MHO in Adolescents

The nutrition intakes of MUO and MHO in adolescents are shown in Table 4. All factors of nutrition intake were not significantly different between adolescents in the MUO group and those in the MHO group.

### 3.8. The Correlation between Metabolites Associated with Clinical Parameters and IR Assessment Index of MUO and MHO in Adolescents

The current study expressed a correlation heatmap containing the metabolites, clinical parameters and IR assessment index of MUO and MHO in adolescents (Figure 1A–D). For example, C3-OH correlated positively with the weight^th^, BMI%, BMI, HC, BF%, fat mass, TGs, T-cholesterol, and TyG index and correlated negatively with height, FFM, RBC, Hb, and Hct (Figure 1A) in the adolescents of the MHO group. In addition, Figure 1A shows that C3-OH significantly positively correlated with BMI%, BMI, HC, BF%, and body fat mass (r = 0.27735, *p* = 0.0167, r = 0.27335, *p* = 0.0184, r = 0.28242, *p* = 0.0148, r = 0.30879, *p* = 0.0074, and r = 0.23006, *p* = 0.0486, respectively) and correlated negatively with RBC and Hb in the adolescents of the MUO group (r = −0.26086, *p* = 0.0248, and r = 0.28790, *p* = 0.0129, respectively). Moreover, Figure 1A shows that C2 significantly positively correlated with DBP (r = 0.23090, *p* = 0.0478, respectively), Lys correlated negatively with HOMA-IR (Figure 1B; r = −0.26173, *p* = 0.0243), and PCaaC34:1 correlated negatively with Plt (Figure 1D; r = −0.23320, *p* = 0.0455) in the adolescents of the MUO group. His correlated positively with AST and ALT in the adolescents of the MHO group (Figure 1B; r = 0.31157, *p* = 0.0069 and r = −0.27001, *p* = 0.0200, respectively). These were inverse correlations in each group.

## 4. Discussion

This study investigated the difference or/and relationship of clinical features and metabolites between MHO and MUO in adolescents. To investigate the clinical features of MetS in obese adolescents, this study divided the participants into MHO and MUO groups, according to the IDF criteria. The results reveal that there were significant differences between the adolescents in the MHO group and those in the MUO group in terms of laboratory measurements, IR assessment index, and lipid profiles. Furthermore, there were significant differences in the three ACs, five AAs, Gln/Glu ratio, three BAs, and two GPLs between the adolescents in the MHO group and those in the MUO group. The results show that several metabolites were associated with the prevalence of MUO in adolescents. Additionally, several metabolites were inversely correlated with MHO in adolescents of the MUO group. Thus, MUO adolescents have metabolic characteristics despite the same obesity.

Despite the same obese adolescents (even nutrition intake was not significantly different), the obesity or MetS diagnosis factors were significantly different between the adolescents of the MHO group and the adolescents of the MUO group. These factors are known to be good biomarkers of obesity and MetS [21]. In the current study, AST and ALT were positively correlated with His in the adolescents of the MHO group, whereas in the adolescents of the MUO group, an inverse correlation was found. Obese individuals or those with T2DM or IR had a high level of AST and ALT [22]. Some research has shown that although levels of AST and ALT are increased [23], His concentrations are lower [24] in patients with CVD. However, research reported that there was an increase in liver AST following an intake of His supplementation by obese women with MetS [25]. Further research is warranted to understand the correlation between AST, ALT, and His in obese or MetS adolescents.

Unlike the HOMA-IR, the TyG index was significantly higher in the adolescents of the MUO group than in the adolescents of the MHO group. Moreover, the OR for MUO prevalence was 2.046-fold higher for higher quartiles of the TyG index. Recently, the TyG index is being considered an IR assessment index along with HOMA-IR [26]. In addition, studies have reported that the TyG index is more appropriate for the diagnosis of T2DM than weight gain [27]. Thus, the TyG index could be a good biomarker of MetS in obese adolescents.

The C2, C3-OH, and C5-M-DC of ACs were significantly different between the adolescents of the MHO group and those of the MUO group. Unlike in adolescents of the MHO group, the C2 was positively correlated with DBP in adolescents of the MUO group. Additionally, the OR for MUO prevalence was 1.606-fold higher for higher quartiles of the C2. ACs have a trend to increase the level of individuals who have a risk of obesity and MetS [28]. ACs are divided by the length of carbon chains in the molecular structure, such as free carnitine; C0, short-chain; C2–C5, medium-chain; C6–C12, and long-chain C14–C18 [29]. C2 of short-chain ACs has a positive correlation with BMI in patients with T2DM [30]. Furthermore, C2 has been associated with SBP [31]. Some research showed that the level of C3-OH was significantly higher in patients with peripheral artery disease, diabetic nephropathy, and DM than in the control group [32,33]. A study showed that C5-M-DC was associated with IR, in mice with DM [34]. In addition, C5-M-DC has a strong relationship between glomerular filtrating rate and creatinine in patients with CVD [35]. Furthermore, the current study investigated the predictors of the MUO adolescent prevalence as ACs by length. As a result of the odds ratio for each length of the AC quartile and the occurrence of MUO, there was a significant difference only in the quartile of short-chain AC (Appendix A). Some research suggested that an increase in short-chain ACs was associated with MetS such as T2DM and obesity [36,37]. Therefore, C2, C3-OH, and C5-M-DC of ACs may be metabolic biomarkers of related MetS in obese adolescents.

The Gln, His, Lys, Ser, and Gln/Glu ratios were significantly lower in the adolescents of the MUO group than in the adolescents of the MHO group. On the contrary, Ala was significantly higher in the adolescents of the MUO group compared with the adolescents of the MHO group. According to a study, levels of Ala were higher in obese individuals than in normal-weight individuals [38]. Furthermore, Ala is a gluconeogenic substrate secreted by skeletal muscle at higher levels in patients with DM [39]. Some studies have suggested that Ser has therapeutic potential for DM [40], such as improved regulation of blood glucose [41] and insulin secretion [42]. Lys levels were decreased in MetS individuals (in particular, cardio-metabolic features and inflammatory biomarkers), as per a study [43]. Another study suggested that Lys has potential protective effects against MetS [44]. These results are in line with the results of our study. Therefore, Ala, Gln, His, Lys, and Ser might be good biomarkers of MetS in obese adolescents. According to Cheng et al. [45], individuals with MetS have increased levels of the Gln/Glu ratio. Moreover, some researchers have suggested that Gln/Glu ratios [46,47] are associated with MetS such as DM, CVD, and IR. In addition, in the present study, the level of Glu was higher in the MUO group than in the MHO group; however, it was not significant (Table 2). Thus, Glu, Gln, and Gln/Glu ratios are useful biomarkers of MetS in adolescents. However, Gln was significantly lower in hyperuricemia patients with MetS [48]. Therefore, detailed studies and research regarding MetS and other diseases in individuals, are required in the future.

The kynurenine, Met-SO, and spermidine were significantly higher in the adolescents of the MUO group than in those of the MHO group. Some researchers reported that high levels of kynurenine are associated with obesity [49], idiopathic pulmonary arterial hypertension [50], and IR [51]. Met-SO was significantly lower in patients with HTN who were on a low-sodium diet [52] and significantly higher in patients with DM [53]. Thus, the results of this study suggest that kynurenine and Met-SO are associated with MetS in obese adolescents. According to Choksomngam et al. [54], in animal models, spermidine supplementation has been shown to protect against diet-induced obesity. Furthermore, in humans [55] and mice [56], spermidine intake correlates with reduced blood pressure and decreased risk of CVD. In the current study, the levels of spermidine were higher in the adolescents of the MUO group than in adolescents of the MHO group. The results of this study are the opposite of the results of some studies. Therefore, further studies are needed to understand the effects of spermidine in MUO and MHO in adolescents.

The results of the present study show that two GPLs were significantly higher in the adolescents of the MUO group than in the adolescents of the MHO group. According to Jové et al. [57], levels of GPLs were higher in the overweight and obese group with MetS than in the overweight and obese group without MetS. Furthermore, GPLs are associated with related MetS factors such as LDL-cholesterol, glucose, and IR in humans [58] and rats [59]. However, for each length or double bond, the GPLs may be different in the individuals with MetS. In this study, PCaaC34:1 as a significant biomarker had a significantly higher prevalence OR of MUO in adolescents as against PCaaC32:2. There is research that PCaaC34:1 reduction in type 1 DM [60], an increase in Alzheimer’s disease [61] and sepsis could occur in the event of community-acquired pneumonia [60]. However, there is insufficient research regarding PCaaC34:1 in individuals with MetS or obesity. Despite the need for further studies of GPLs in adolescents with MetS, the results of this study suggest that GPLs have the potential as MetS biomarkers in obese adolescents.

In particular, this study focused on two results. First, MetS associated an increase in short-chain ACs in obese adolescents. Second, the correlation results of Lys (with HOMA-IR) and PCaaC34:1 (with Plt) in adolescents in the MUO group and His (with AST and ALT) correlation results in adolescents in the MHO group. Furthermore, the TyG index also indicated a deeper relationship with the adolescents in the MUO group than with the adolescents in the MHO group. By showing differences from the results of the MHO group, these results suggest biomarkers associated with MetS in obese adolescents. However, this study had certain limitations. Due to the cross-sectional design of the study, it was difficult to establish a causal relationship between metabolites and MetS in adolescents. Another limitation of this study is that it was conducted with a small number of Korean adolescents aged 14–16 years. Hence, the application of general metabolite properties for MetS obtained in the present study may be limited in adolescents of other ages, such as preschool children or adolescents aged 17 years and older, and could not be a population of all races. Therefore, further studies using large sample sizes or more varied races are required to investigate the interaction between metabolites and MetS in adolescents.

Nevertheless, the strength of this study is that it thoroughly measured and analyzed various clinical and metabolic parameters of the participants, and investigated independent predictors of the odds ratio and the presence or absence of MetS. Furthermore, the study’s findings add to the evidence of the potential use of these biomarkers to identify and characterize MUO adolescents. Obesity in children and adolescents is caused by an imbalance in energy intake and physiological activity [62], or a complex factor of environment and genetics [63], which is why comprehensive and in-depth research on metabolic syndrome caused by obesity is necessary. Metabolites are useful biomarkers that reflect the clinical state of these metabolic syndromes [64,65]. That is the reason why biomarker discovery will play an essential role in the prediction, diagnosis, and management of metabolic syndrome in the future. It is important to define obesity using biochemical factors in addition to anthropometric and clinical factors. As shown in the results of this study, clinically, the same obesity clearly exists as healthy obesity and unhealthy obesity, but the differences were seen in biomarkers predicting metabolic syndrome. In other words, several metabolites found in this study will function as predictive biomarkers for being metabolically unhealthy in obese adolescents.

Ultimately, the present study suggests that several metabolite biomarkers of MetS found in this study are characteristic of MUO adolescents. Additionally, these metabolites and the TyG index proved to be excellent biomarkers reflecting the presence of MetS clinical symptoms and IR-related diseases. Consequently, the results of this study will be a reliable reference for specific biomarkers related to metabolic syndrome in MUO adolescents.

## 5. Conclusions

The current study found significant differences in certain AAs, ACs, BAs, GPLs, and the Gln/Glu ratio between the MUO and MHO groups. In addition, the study observed that His, Lys, PCaaC34:1, and several clinical factors in adolescents of the MUO group were reverse correlated with the results in adolescents of the MHO group. The TyG index also was related to MUO in adolescents compared with HOMA-IR. In particular, it was associated with C2, Ala, Glu, Gln/Glu ratio, kynurenine, Met-SO, C34:1, HOMA-IR, and TyG index in MUO adolescents. Therefore, these biomarkers found in this study have the potential to reflect the clinical outcomes of MUO in adolescents. These biomarkers will lead to a better understanding of the specificity and properties of MetS in obese adolescents.

## Figures and Tables

**Figure 1 children-10-00322-f001:**
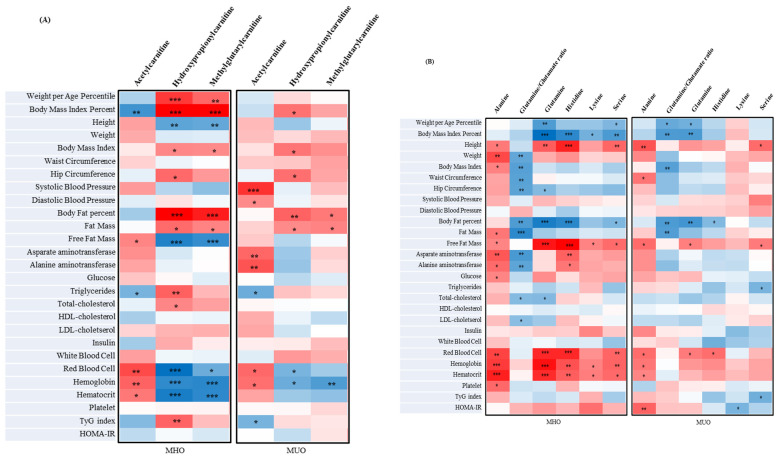
Correlation heatmap between metabolites associated with clinical parameters and IR assessment index in MHO and MUO. Correlation heatmap between 3 acylcarnitines (**A**) and 6 amino acids (**B**) associated with anthropometric parameters, and laboratory measurements in metabolically healthy obesity (MHO) and metabolically unhealthy obesity (MUO) groups. Partial correlation coefficient at the MHO and MUO. ** p* < 0.05, *** p* < 0.01, and **** p* < 0.001. Correlation heatmap between 3 biogenic amines (**C**) and 2 glycerophospholipids (**D**) associated with anthropometric parameters, and laboratory measurements in metabolically healthy obesity (MHO) and metabolically unhealthy obesity (MUO) groups. Partial correlation coefficient at the MHO and MUO. ** p* < 0.05, *** p* < 0.01, and **** p* < 0.001 PC, phosphatidylcholine; aa, diacyl.

**Table 1 children-10-00322-t001:** Anthropometric parameters, characteristics, laboratory measurements, IR assessment index, and plasma lipid profiles of MUO and MHO in adolescents.

		Total Participants (*n* = 148)				
		MHO (*n* = 74)		MUO (*n* = 74)		*p*
**Anthropometric and Characteristics** * ^∮^ *						
Male/Female	*n*, (%)	82 (55.41)/66 (44.59)				
		36 (48.65)/38 (51.35)		46 (62.16)/28 (37.84)		0.0982
Age		14.08	±0.097	13.93	±0.089	0.2625
Height	cm	163.7	±0.809	166.9	±0.944	0.0148
Weight	kg	91.12	±1.267	95.19	±1.816	0.0983
Body Mass Index	kg/m^2^	33.92	±0.329	34.06	±0.453	0.9531
Waist Circumference	cm	102.3	±0.912	104.6	±1.113	0.1402
Hip Circumference	cm	112.6	±0.616	113.2	±0.925	0.6343
Systolic Blood Pressure	mmHg	118.2	±1.168	133.7	±1.453	<0.0001
Diastolic Blood Pressure	mmHg	75.34	±0.772	84.19	±1.050	<0.0001
Body fat	%	45.42	±0.626	44.22	±0.741	0.1739
Fat Mass	kg	41.56	±0.966	42.43	±1.239	0.8330
Free Fat Mass	kg	49.57	±0.760	52.76	±1.014	0.0169
BMI percentage	%	99.59	±0.039	99.23	±0.146	0.0164
Weight per age percentile	*n* ^th^	99.97	±0.004	99.77	±0.073	0.0073
Basal Metabolic Rate	Kcal	1169.3	±13.28	1214.8	±18.52	0.0480
**Laboratory measurements** * ^∮^ *						
Asparate aminotransferase	IU/L	27.68	±2.089	29.57	±1.866	0.4439
Alanine aminotransferase	IU/L	36.81	±4.257	43.28	±4.336	0.1794
White Blood Cell	×10^3^/mm^3^	7.013	±0.202	7.485	±0.164	0.0490
Red Blood Cell	×10^3^/mm^3^	4.949	±0.036	5.043	±0.034	0.1841
Hemoglobin	g/dL	14.19	±0.115	14.38	±0.116	0.6055
Hematocrit	g/dL	43.59	±0.309	44.20	±0.314	0.3160
Platele	×10^3^/mm^3^	311.9	±7.332	326.6	±5.733	0.0714
Insulin	*n*g/mL	27.06	±2.697	29.53	±1.944	0.2221
Glucose	mg/dL	92.50	±0.744	98.65	±2.439	0.0146
**Insulin resistance (IR) assessment index** * ^∮^ *						
Homeostasis model assessment of insulin resistance (HOMA-IR)		6.218	±0.630	7.389	±0.616	0.2705
Triglyceride-Glucose Index (TyG index)		8.417	±0.052	8.792	±0.059	<0.0001
**Plasma lipid profiles** * ^∮^ *						
Triglycerides	mg/dL	107.1	±5.358	150.4	±8.202	<0.0001
Total cholesterol	mg/dL	173.5	±3.119	180.5	±3.416	0.0878
LDL-cholesterol	mg/dL	106.2	±2.834	105.9	±3.071	0.9564
HDL-cholesterol	mg/dL	45.84	±0.657	44.51	±1.167	0.1281

Mean ± SE. *^∮^* tested in the following logarithmic transformation, *p*-values were derived from a general linear model with a Bonferroni correction adjusting for age, sex and body mass index of the adolescents in the metabolically unhealthy obesity (MUO) group and of the adolescents in the metabolically healthy obesity (MHO) group.

**Table 2 children-10-00322-t002:** The metabolites of MUO and MHO in adolescents.

		Total Participants (*n* = 148)				
		MHO (*n* = 74)		MUO (*n* = 74)		*p*
**Acylcarnitines** * ^∮^ *						
Acetylcarnitine	C2	8.486	±0.371	9.796	±0.362	0.0184
Hydroxypropionylcarnitine	C3-OH	0.072	±0.004	0.077	±0.003	<0.0001
Methylglutarylcarnitine	C5-M-DC	0.033	±0.001	0.035	±0.001	0.0273
**Amino Acids** * ^∮^ *						
Alanine		421.3	±10.45	462.3	±80.65	0.0125
Glutamine		569.4	±13.19	545.1	±13.38	0.0041
Glutamate		108.3	±4.889	120.0	±4.877	0.1244
Glutamine/Glutamate ratio		5.926	±0.252	5.098	±0.230	0.0179
Histidine		94.64	±1.645	92.43	±1.539	0.0322
Lysine		229.7	±4.593	218.9	±4.463	0.0371
Serine		134.6	±2.921	126.9	±2.714	0.0061
**Biogenic Amines** * ^∮^ *						
Kynurenine		2.168	±0.073	2.612	±0.086	0.0003
Methionine-sulfoxide		0.657	±0.028	0.761	±0.026	0.0194
Spermidine		0.202	±0.015	0.236	±0.012	0.0346
**Glycerophospholipids** * ^∮^ *						
PC aa C32:2		2.901	±0.109	3.180	±0.117	0.0397
PC aa C34:1		163.1	±4.379	179.3	±4.703	0.0402

Mean ± SE. *^∮^* tested in the following logarithmic transformation, *p*-values were derived from a general linear model with a Bonferroni correction adjusting for age, sex and body mass index (BMI) of the adolescents in the metabolically unhealthy obesity (MUO) group and of the adolescents in the metabolically healthy obesity (MHO) group. PC, phosphatidylcholine; aa, diacyl.

**Table 3 children-10-00322-t003:** Predictors of the MUO adolescents’ prevalence ORs on significantly different metabolites.

Variables		Total Subjects (*n* = 148)	*p*
		ORs (95% CI) For MUO Adolescents	
**Quartile of Acylcarnitines**			
Acetylcarnitine	C2	1.606 (1.191–2.165)	0.0014
Hydroxypropionylcarnitine	C3-OH	1.114 (0.835–1.488)	0.4620
Methylglutarylcarnitine	C5-M-DC	1.114 (0.835–1.488)	0.4620
**Quartile of amino acids**			
Alanine		1.621 (1.194–2.200)	0.0014
Glutamine		0.833 (0.624–1.112)	0.2131
Glutamate		1.364 (1.024–1.818)	0.0316
Glutamine/Glutamate ratio		0.735 (0.547–0.988)	0.0388
Histidine		0.948 (0.712–1.263)	0.7146
Lysine		0.839 (0.626–1.123)	0.2363
Serine		0.796 (0.599–1.058)	0.1142
**Quartile of biogenic amines**			
Kynurenine		1.661 (1.222–2.259)	0.0008
Methionine-sulfoxide		1.983 (1.371–2.867)	0.0001
Spermidine		1.304 (0.971–1.751)	0.0752
**Quartile of glycerophospholipids**			
PC aa C32:2		1.258 (0.939–1.685)	0.1214
PC aa C34:1		1.424 (1.057–1.917)	0.0180
**Quartile of insulin resistance assessment index**			
Homeostasis model assessment of insulin resistance (HOMA-IR)		1.360 (1.012–1.827)	0.0388
TyG index		2.046 (1.476–2.835)	<0.0001

*p*-values derived from a logistic regression analysis on the metabolically unhealthy obesity (MUO).

**Table 4 children-10-00322-t004:** The nutrition intakes of adolescents in MUO and MHO groups.

		Total Participants (*n* = 148)				
		MHO (*n* = 74)		MUO (*n* = 74)		*p*
**Nutrition Intakes** * ^∮^ *						
Energy	Kcal	1424.7	±45.72	1523.0	±40.64	0.0600
Carbohydrate	g	211.3	±6.588	229.1	±6.650	0.0827
Protein	g	56.44	±2.117	60.16	±1.751	0.0698
Fat	g	36.93	±1.801	38.24	±1.309	0.2151
Total fiber	g	9.163	±0.459	9.922	±0.459	0.2348
Soluble fiber	g	1.798	±0.129	2.020	±0.119	0.1700
Non-soluble fiber	g	6.703	±0.352	7.406	±0.353	0.1336
Cholesterol	mg	174.5	±10.41	186.0	±11.75	0.3985
Calcium	mg	366.7	±19.89	404.1	±17.10	0.0677
Potassium	mg	1615.1	±57.48	1776.7	±61.32	0.0579
Sodium	mg	2529.6	±100.3	2692.9	±111.4	0.3018
Total amino acids	mg	31,383.4	±1340.8	32,567.8	±1561.1	0.5081
Essencial amino acids	mg	14,294.0	±627.4	14,950.5	±744.5	0.4711
Non-essencial amino acids	mg	17,089.4	±823.5	17,617.3	±822.1	0.5450
Isoleucine	mg	1333.5	±59.16	1378.2	±67.69	0.4235
Leucine	mg	2533.9	±110.7	2661.2	±133.3	0.5552
Valine	mg	1554.5	±71.39	1621.5	±77.19	0.4277
Glutamic acid	mg	6212.1	±279.1	6319.9	±289.8	0.5778
Total fatty acids	g	28.75	±1.504	28.75	±1.227	0.6784
Total trans-fatty acids	g	0.368	±0.026	0.360	±0.018	0.4523
Total essential fatty acids	g	7.836	±0.350	8.425	±0.436	0.4208
Total saturated fatty acids	g	9.922	±0.669	9.702	±0.438	0.5272
Total mono-unsaturated fatty aicds	g	10.201	±0.628	9.902	±0.455	0.8554
Total poly-unsaturated fatty acids	g	8.260	±0.363	8.788	±0.451	0.5087
*n*-3 fatty acids	g	0.861	±0.044	0.952	±0.057	0.2989
*n*-6 fatty acids	g	7.161	±0.314	7.666	±0.388	0.4401

Mean ± SE. *^∮^* tested in the following logarithmic transformation, *p*-values were derived from a general linear model with a Bonferroni correction adjusting for age, sex and body mass index of the adolescents in the metabolically unhealthy obesity (MUO) group and of the adolescents in the metabolically healthy obesity (MHO) group.

## Data Availability

All data generated or analyzed during this study are included in this article. The datasets generated and/or analyzed during the current study are not publicly available due [reason for research ethics] but are available from the corresponding author on reasonable request.

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
