# Peer review of "Identification of Biomarkers Related to Metabolically Unhealthy Obesity in Korean Obese Adolescents: A Cross-Sectional Study"

_children, 2023, doi:10.3390/children10020322_

Round 1
Reviewer 1 Report
The manuscript presented by Jeong et al is a well-written original paper. The authors describe clearly the material and methods and results.
However, some modifications are needed.
In the results sections, I suggest re-organizing the tables after every paragraph, to be easier to follow the results.
Please rewrite the following paragraph in a more explicit way:
3.7. Predictors of the MUO adolescents’ prevalence ORs on significantly different metabolites
Predictors of the MUO adolescents’ prevalence ORs on significantly different metabolites are shown in Table 3. The ORs of the adolescents in the MUO group, obtained using a logistic regression analysis showed that the prevalence ORs of the C2, Ala, Gln/Glu ratio, kynurenine, Met-SO, C34:1, HOMA-IR, and TyG index were significantly greater than ORs of the MUO occurrence, compared with each quartile of the metabolites.
I suggest the authors be more focused on the discussions section, and on the importance of their research. Any of these metabolites can be used on predicting MUO development, maybe using an algorithm in the future? How these metabolites might offer better solutions in terms of early diagnosis of MUO.
Conclusions can be improved to be more specific.
Author Response
Dear Editor in Chief,
Thank you for your helpful comments and concerns that allowed us to improve the quality of the manuscript, greatly. We agree with all your comments, and we have corrected the manuscript accordingly, point-by-point.
Reviewer #1:
Comments and Suggestions for Authors
The manuscript presented by Jeong et al is a well-written original paper. The authors describe clearly the material and methods and results.
However, some modifications are needed.
In the results sections, I suggest re-organizing the tables after every paragraph, to be easier to follow the results.
Author’s response: We apologize for the confusion and thank you for your careful comments.
As per your comments, we have revised the order of results subsection in the ‘3. Results’ section as follows:
3.4. IR assessment index of MUO and MHO in adolescents
3.5. The metabolites of MUO and MHO in adolescents
3.6. Predictors of the MUO adolescents’ prevalence ORs on significantly different metabolites
3.7. The nutrition intakes of MUO and MHO in adolescents
3.8. The correlation between metabolites associated with clinical parameters and IR assessment index of MUO and MHO in adolescents
Please rewrite the following paragraph in a more explicit way:
3.7. Predictors of the MUO adolescents’ prevalence ORs on significantly different metabolites
Predictors of the MUO adolescents’ prevalence ORs on significantly different metabolites are shown in Table 3. The ORs of the adolescents in the MUO group, obtained using a logistic regression analysis showed that the prevalence ORs of the C2, Ala, Gln/Glu ratio, kynurenine, Met-SO, C34:1, HOMA-IR, and TyG index were significantly greater than ORs of the MUO occurrence, compared with each quartile of the metabolites.
Author’s response: Thank you for your valuable suggestions.
We conducted multivariate logistic regressions to analyze predictors of the MUO adolescents' prevalence odds ratio on metabolites. Consequently, we have shown values of the odds ratio and confidence interval on each metabolite.
As per your suggestion, we revised in the ‘3.7. Predictors of the MUO adolescents’ prevalence ORs on significantly different metabolites’ as follow:
(Line 252–262 in the revised manuscript):
3.6. Predictors of the MUO adolescents’ prevalence ORs on significantly different metabolites
Predictors of the MUO adolescents’ prevalence ORs on significantly different metabolites are shown in Table 3. ORs for MUO prevalence were each fold higher in the upper quartiles of the C2 [OR; 1.606; 95% confidence interval (CI): 1.191–2.165; P = 0.0014], Ala [OR; 1.621; 95% CI: 1.194–2.200; P = 0.0014], Glu [OR; 1.364; 95% CI: 1.024–1.818; P = 0.0316], Gln/Glu ratio [OR; 0.735; 95% CI: 0.547–0.988; P = 0.0388], kynurenine [OR; 1.661; 95% CI: 1.222–2.259; P = 0.0008], Met-SO [OR; 1.983; 95% CI: 1.371–2.867; P = 0.0001], C34:1 [OR; 1.424; 95% CI: 1.057–1.917; P = 0.0180], HOMA-IR [OR; 1.360; 95% CI: 1.012–1.827; P = 0.0388], and TyG index [OR; 2.046; 95% CI: 1.476–2.835; P<0.0001]. On the other hand, every other metabolite, except for these metabolites, showed no significant difference between the adolescents in the MUO group and the adolescents in the MHO group.
I suggest the authors be more focused on the discussions section, and on the importance of their research. Any of these metabolites can be used on predicting MUO development, maybe using an algorithm in the future? How these metabolites might offer better solutions in terms of early diagnosis of MUO.
Author's response: We appreciate your concern comments.
Obesity in children and adolescents is caused by an imbalance in energy intake and physiological activity or a combination of environmental and genetic factors, so comprehensive and in-depth research on metabolic syndrome caused by obesity is needed. Metabolites are useful biomarkers reflecting the clinical status of these metabolic syndromes. This is why biomarker discovery will play an essential role in predicting, diagnosing, and managing metabolic syndrome in the future. Thus, the results of this study were evaluated as reliable biomarkers for predicting metabolically unhealthy in obese adolescents.
As a result of this study, there were significant differences between the metabolically healthy obesity (MHO) group and the metabolically unhealthy obesity (MUO) group in terms of the metabolites, laboratory measurement, insulin resistance assessment index, and lipid profiles. In particular, we found references related to metabolic syndromes such as diabetes melitus, cardiovascular disease, and obesity related to these biomarkers observed in this study. Therefore, we considered these biomarkers to be potentially valuable for identifying and characterizing obese adolescents with metabolic syndrome risk factors.
As per your suggestion, we have some text added and revised in the ‘4. Discussion’ section as follows:
(Line 478–493 in the revised manuscript):
Nevertheless, the strength of this study is that it thoroughly measured and analyzed various clinical and metabolic parameters of the participants, and investigated independent predictors of the odds ratio and the presence or absence of MetS. Furthermore, the study's findings add to the evidence of the potential use of these biomarkers to identify and characterize MUO adolescents. Obesity in children and adolescents is caused by an imbalance in energy intake and physiological activity [62], or a complex factor of environment and genetics [63], which is why comprehensive and in-depth research on metabolic syndrome caused by obesity is necessary. Metabolites are useful biomarkers that reflect the clinical state of these metabolic syndromes [64, 65]. That is the reason why biomarker discovery will act an essential role in the prediction, diagnosis, and management of metabolic syndrome in the future. It is very important to define obesity using clinical factors in addition to anthropological factors. As shown in the results of this study, clinically, the same obesity clearly exists as healthy obesity and unhealthy obesity, but the differences were seen in biomarkers predicting metabolic syndrome. In other words, several metabolites found in this study will function as predictive biomarkers of metabolically unhealthy in obese adolescents.
Conclusions can be improved to be more specific.
Author's response: Thank you for your careful concern.
We found clinical factors and several metabolites specific to MUO adolescents whereas MHO adolescents. As a result of this study, it could suggest identifying MUO adolescents. Furthermore, this study will help as a reference to confirm the potential risk of metabolic syndrome in obese adolescents. So, we suggested in the 4. Discussion of a manuscript on the potential clinical implications of these metabolites on metabolic syndrome. Also, your comment is very important. Therefore, we have added more detailed text about the ‘5. Conclusions’ in the manuscript.
As per your suggestion, we have added text in the '5. Conclusions' section as follows:
(Line 502–511 in the revised manuscript):
5. Conclusions
The current study found significant differences in certain AAs, ACs, BAs, GPLs, and the Gln/Glu ratio between the MUO and MHO groups. In addition, the study observed that His, Lys, PCaaC34:1, and several clinical factors in adolescents of the MUO group were reverse correlated with the results in adolescents of the MHO group. The TyG index also was related to MUO in adolescents compared with HOMA-IR. In particular, it was associated with C2, Ala, Glu, Gln/Glu ratio, kynurenine, Met-SO, C34:1, HOMA-IR, and TyG index in MUO adolescents. Therefore, these biomarkers found in this study have the potential to reflect the clinical outcomes of MUO in adolescents. These biomarkers will lead to a better understanding of the specificity and properties of MetS in obese adolescents.
Submission Date
26 December 2022
Date of this review
08 Jan 2023 15:06:13
Thank you again for your profound advice, and we hope that our revision meets your requirements for publication as a final revision.
Sincerely yours,
Hye-Ja Lee, Ph.D.
Division of Endocrine and Kidney Disease Research, Department of Chronic Disease Convergence Research, Korea National Institute of Health, Korea Disease Control and Prevention Agency, 187 Osongsaengmyeong 2-ro, Osong-eup, Cheongju-si, Chungcheongbuk-do, 28159, Republic of Korea
Fax: +82-43-719-8709, Tel: +82-43-719-8452, E-mail: hyejalee@yahoo.co.kr

Reviewer 2 Report
The study by Sarang Jeong et al. presents a cross-sectional study that aimed to identify potential metabolic biomarkers related to metabolically unhealthy obesity (MUO) in Korean obese adolescents. A total of 148 adolescents were divided into MUO and metabolically healthy obesity (MHO) groups based on age-specific adolescent metabolic syndrome criteria of the International Diabetes Federation. The study found significant differences in certain amino acids, acylcarnitines, biogenic amines, glycerophospholipids, and the glutamine/glutamate ratio between the MUO and MHO groups. These biomarkers were also found to be associated with the prevalence of MUO and inversely correlated with MHO in the MUO group.
One of the strengths of the paper is the thorough measurement and analysis of various clinical and metabolic parameters in the study participants. The use of multivariate analysis to investigate the independent predictors for the odds ratio and presence of metabolic syndrome is also a strength. The study's findings add to the growing body of evidence on the potential use of these biomarkers in identifying and characterizing metabolically unhealthy obesity in adolescents.
Major comments:
1. One major concern with the study is the lack of description of the recruitment process and the representativeness of the sample. It is not clear how the participants were recruited or whether the sample is representative of the general population of Korean adolescents with obesity. This makes it difficult to determine the external validity of the study and to generalize the findings to other populations.
Minor comments:
1. It would be helpful to provide more information on the selection criteria for the study participants. How were the adolescents recruited for the study and how were they screened for inclusion?
2. The sample size of 148 adolescents may not be large enough to draw robust conclusions about the prevalence and relationships of the biomarkers identified in the study. It would be useful to see the power calculation for the sample size and whether the sample size was sufficient to detect the differences observed in the study.
3. It would be helpful to provide more detailed information about the dietary intake assessment methods used in the study, such as how the 3-day food records were validated and what formulas were used to calculate averages of energy, macronutrients, and micronutrients.
4. It would be useful to provide more information on the statistical methods used in the analysis, such as the type of regression analysis conducted.
5. The study did not provide information on the physical activity levels of the participants. These factors could potentially affect the metabolic profiles and the development of metabolically unhealthy obesity in adolescents.
6. In the results section, it would be helpful to provide the actual values for the metabolite concentrations rather than just stating that they are significantly different between the MUO and MHO groups.
7. Finally, it would be helpful if the authors provided more information about the potential clinical implications of their findings. It is not clear how the identified biomarkers may be used in practice to identify and manage MUO in adolescents.
Overall, the study investigates an important and timely topic and the results provide valuable insights into the differences between metabolically healthy and unhealthy obesity in adolescents. However, some additional context and clarification in the manuscript would greatly improve its readability and overall impact.
Thank you for considering my comments.
Author Response
Reviewer #2:
Summary
The manuscript by Jeong and Lee evaluated the reliability of a triglyceride-glucose (TyG) index as an insulin resistance assessment marker. The study included 378 overweight participants that were placed in tertiles according to HOMA-IR and the TyG index and investigated their relationship with numerous IR-related factors. Although the study investigates an interesting and important topic, there are some issues to address.
Comments and Suggestions for Authors
The study by Sarang Jeong et al. presents a cross-sectional study that aimed to identify potential metabolic biomarkers related to metabolically unhealthy obesity (MUO) in Korean obese adolescents. A total of 148 adolescents were divided into MUO and metabolically healthy obesity (MHO) groups based on age-specific adolescent metabolic syndrome criteria of the International Diabetes Federation. The study found significant differences in certain amino acids, acylcarnitines, biogenic amines, glycerophospholipids, and the glutamine/glutamate ratio between the MUO and MHO groups. These biomarkers were also found to be associated with the prevalence of MUO and inversely correlated with MHO in the MUO group.
One of the strengths of the paper is the thorough measurement and analysis of various clinical and metabolic parameters in the study participants. The use of multivariate analysis to investigate the independent predictors for the odds ratio and presence of metabolic syndrome is also a strength. The study's findings add to the growing body of evidence on the potential use of these biomarkers in identifying and characterizing metabolically unhealthy obesity in adolescents.
Major comments:
One major concern with the study is the lack of description of the recruitment process and the representativeness of the sample. It is not clear how the participants were recruited or whether the sample is representative of the general population of Korean adolescents with obesity. This makes it difficult to determine the external validity of the study and to generalize the findings to other populations.
Author's response: We appreciate your comments.
Based on your comments, we focus on the argument of non-representational in cross-sectional studies. This study was conducted as a cross-sectional study with 148 adolescents aged 14-16 among the subjects recruited to the Korean Children-Adolescents Study (KoCAS). In addition, among the 148 obese adolescents, 143 were severely obese with BMIs ≥ 99th percentile (or ≥ 30 kg/m2). We are also aware that your comment is one of this study's limitations. As a result of this study, only the baseline participants among the cohort study participants were analyzed. Nevertheless, this study calculated the sample size of the participants in this study by applying a 95% confidence interval and an 8% margin of error to the 2012 Korean adolescent population. Furthermore, we additionally analyzed and compared on characteristics of MHO and MUO in adolescents with the current study using Korea National Health and Nutritional Examination Survey (KNHANES) in 2012. The additional analysis is as follows:
Although 79 were only obese adolescents, like the results of the current study, the mean of systolic blood pressure, total cholesterol, glucose, and triglycerides in the MUO group was significantly higher than that in the MHO group. The mean high-density lipoprotein cholesterol in the MUO group was significantly lower than that in the MHO group. These results make interpreted as similar results with our study. Like the above your comment, our study participants are a non-representative population. However, the current study participants have higher obesity than participants of KNHANES. So, we can analyze more many MUO than KNHANES. Although some characteristics of 79 obese adolescents were presented using only the 2012 KNHANES, the characteristics of Korean adolescents can be understood considering that KNHANES is a sample of Korean adolescents. So, we added some text in the ‘Study participants in the 2. Materials and Methods’ and this limitation was added to the text in the ‘4. Discussion’ section as follows:
(Line 82–89 in the revised manuscript):
2. Materials and Methods
Study participants
The data were obtained from the Korean Children-Adolescents Study (KoCAS), conducted by the Korean National Institute of Health. The KoCAS subjects in this study were adolescents aged 14–16 years from Seoul and Gyeonggi provinces for whom clinical biomarker data were collected 2012. Obesity was defined as a body mass index (BMI) > 25 kg/m2 or being in the 95th percentile for age and sex according to the 2017 Korean growth standard [2] for children and adolescents. Among the adolescents, 143 who were severely obese adolescents aged 14-16 years with BMIs ≥ 99th percentile (or ≥ 30 kg/m2) were used in present study.
(Line 471–475 in the revised manuscript):
Another limitation of this study is that it was conducted with a small number of Korean adolescents aged 14–16 years. Hence, the application of general metabolite properties for MetS obtained in the present study may be limited in adolescents of other ages, such as preschool children or adolescents aged 17 years and older, and couldn't be a population of all races.
Minor comments:
It would be helpful to provide more information on the selection criteria for the study participants. How were the adolescents recruited for the study and how were they screened for inclusion?
Author's response: Thank you for your helpful comments.
We agree with your comments. As per your suggestion, we have revised some text in the ‘Study participants in the 2. Materials and Methods’ section as follows:
(Line 82–89 in the revised manuscript):
2. Materials and Methods
Study participants
The data were obtained from the Korean Children-Adolescents Study (KoCAS), conducted by the Korean National Institute of Health. The KoCAS subjects in this study were adolescents aged 14–16 years from Seoul and Gyeonggi provinces for whom clinical biomarker data were collected 2012. Obesity was defined as a body mass index (BMI) > 25 kg/m2 or being in the 95th percentile for age and sex according to the 2017 Korean growth standard [2] for children and adolescents. Among the adolescents, 143 who were severely obese adolescents aged 14-16 years with BMIs ≥ 99th percentile (or ≥ 30 kg/m2) were used in present study.
2. The sample size of 148 adolescents may not be large enough to draw robust conclusions about the prevalence and relationships of the biomarkers identified in the study. It would be useful to see the power calculation for the sample size and whether the sample size was sufficient to detect the differences observed in the study.
Author's response: Thank you for your important comments.
Like the above response to your major comment, we calculated about the 95% confidence interval and 8% margin of error on populations, finally getting about 150 adolescents. So, we added to the text in the ‘4. Discussion’ section as follows:
(Line 471–475 in the revised manuscript):
Another limitation of this study is that it was conducted with a small number of Korean adolescents aged 14–16 years. Hence, the application of general metabolite properties for MetS obtained in the present study may be limited in adolescents of other ages, such as preschool children or adolescents aged 17 years and older, and couldn't be a population of all races.
3. It would be helpful to provide more detailed information about the dietary intake assessment methods used in the study, such as how the 3-day food records were validated and what formulas were used to calculate averages of energy, macronutrients, and micronutrients.
Author's response: Thank you for your valuable comments.
In this study, participants were asked to record their meal and snack intake, including beverages, over a non-consecutive period of 3 days in a week with 1 weekend and 2 weekdays. Ingredients and portions of meals were also recorded. To increase reporting accuracy, written 3-day food records were reviewed and verified using food models in face-to-face interviews by well-trained dieticians. Average values of energy (Kcal), macronutrients (g), and micronutrients (mg) were calculated using research-based representative food intake data and analyzed through the National Health and Nutrition Survey food consumption survey database. So, we explained some sentences in the ‘Assessment of nutrition intake in the 2. Materials and Methods’ section as follows:
(Line 145–156 in the revised manuscript):
Assessment of nutrition intake
In this study, participants were interviewed to obtain information about their nutrient intake. The dietary intake was assessed using a 3-day food record method. The data of food records were validated through their parents or caregiver. The study participants were asked to record their intake of meals and snacks, including beverages during a nonconsecutive period of 3 days (1 day of the weekend and 2 days of the weekdays). Furthermore, ingredients of meals, as well as portion sizes, were also recorded. A well-trained dietician reviewed and confirmed the written 3-day food record using food models during face-to-face interviews to increase precision in reporting. The research-based typical food intake data were used to calculate averages of energy, macronutrients, and micronutrients were analyzed through a food consumption survey database of the Korean National Health and Nutrition Examination Survey.
4. It would be useful to provide more information on the statistical methods used in the analysis, such as the type of regression analysis conducted.
Author's response: Thank you for your careful concern.
As per your suggestion, we have revised some text in the ‘Data processing and statistical analysis in the 2. Materials and Methods’ as follows:
(Line 190 in the revised manuscript):
Data processing and statistical analysis
The statistical analyses were conducted using the Statistical Analysis System software version 9.4 (SAS Institute, Cary, NC, USA). The skewed variables were transformed logarithmically. For the comparison of categorical variables, a chi-squared test was conducted. The general linear model with a Bonferroni correction adjusting for confounding factors was used to compare the parameters collected from each group. Multivariate logistic regressions were conducted to investigate the metabolites as independent predictors for the odds ratio (OR) and the presence of the MetS in adolescents.
5. The study did not provide information on the physical activity levels of the participants. These factors could potentially affect the metabolic profiles and the development of metabolically unhealthy obesity in adolescents.
Author's response: Thank you for your helpful opinions.
We agree with you. So, we further analyzed the basal metabolic rate (BMR) using the Harris-Benedict formula. As a result, the average BMR in the MUO group was significantly higher than that in the MHO group (P = 0.048). It is considered to be due to the application of the Harris-Benedict formula, which includes height and weight. We have revised some values and text in the 'Anthropometric parameters in the 2. Materials and Methods’, ‘3.2. Anthropometric parameters, characteristics, and laboratory measurements of MUO and MHO in adolescents in the 3. Results’, and ‘Table 1.’ section as follows:
(Line 118–120 in the revised manuscript):
Anthropometric parameters
The anthropometric parameters of the study participants were measured in absence of clothing and shoes in the morning after overnight fasting for 12 h. The body weight and body fat percentage (BF%) were measured using a body composition analyzer (BC418; Tanita, Tokyo, Japan). The height was measured to the nearest 0.1 cm with a wall-mounted stadiometer (DS-102; Jenix, Seoul, Korea). The BMI was calculated as weight in kilograms divided by height in meters squared (kg/m2). The BMI z-score was calculated by BMI-for-age percentiles together with the Lambda-Mu-Sigma method of Cole and Green, which provides a way of obtaining normalized growth percentile standards based on The 2017 Korean National Growth Charts for Children and Adolescents. The waist circumference (WC) was measured, with participants standing straight, using a flexible tapeline at the midpoint of the lower rib and the iliac crest to the nearest 0.1 cm. The hip circumference (HC) was measured using a flexible tapeline at the horizontal circumference of the highest point of the buttocks to the nearest 0.1 cm. The waist-hip ratio was calculated based on recorded WC and HC measurements. The systolic blood pressure (SBP) and diastolic blood pressure (DBP) were measured twice on the arm of participants in a seated position following a rest of at least 5 min using an automatic sphygmomanometer (HEM-907, OMRON Healthcare Co., Kyoto, Japan); the two measurements were then averaged. The basal metabolic rate (BMR; Kcal) was calculated using the equation of the Harris-Benedict formula: BMR = [66.47 (for males) or 65.51(for females)] + [9.6 × weight (kg)] + [1.8 × height (cm)] – [4.7 × age].
(Line 209-213 in the revised manuscript):
3.2. Anthropometric parameters, characteristics, and laboratory measurements of MUO and MHO in adolescents
The anthropometric parameters, characteristics, and laboratory measurements of MUO and MHO in adolescents are shown in Table 1. Sex was not significantly different between the MUO and MHO groups. Weight, BMI, WC, HC, body fat in the MUO group was higher than in the MHO group. However, the difference was not significant. The mean of height, SBP, DBP, fat-free mass (FFM), BMI percentage (BMI%), weight per age percentile (weightth), and BMR in the MUO group was significantly higher than in the MHO group (P = 0.0148, P < 0.0001, P < 0.0001, P = 0.0169, P = 0.0164, P = 0.0073, and P = 0.0480, respectively). The mean value of AST, ALT, red blood cell (RBC), hemoglobin (Hb), hematocrit (Hct), platelet (Plt), and insulin was not significantly different between the two groups. The white blood cell count and glucose levels in the MUO group were significantly higher than in the MHO group; however, these levels were in a normal range for adolescents. We adjusted for factors such as age, sex, and BMI variable, except for self-correction.
(Table 1. in the revised manuscript):
6. In the results section, it would be helpful to provide the actual values for the metabolite concentrations rather than just stating that they are significantly different between the MUO and MHO groups.
Author's response: Thank you for your concern and important comments.
However, we decline to comment on your concerns. As per your comment, we showed the mean of metabolite concentrations in the table. The reason why the metabolite concentration was not shown in the manuscript results is that this study was divided into MUO and MHO, but clinically healthy adolescents were included.
As per your suggestion, we suggested in the ‘Table 2.’ section as follow:
(Table 2. in the revised manuscript):
7. Finally, it would be helpful if the authors provided more information about the potential clinical implications of their findings. It is not clear how the identified biomarkers may be used in practice to identify and manage MUO in adolescents.
Author's response: Thank you for your helpful and important comments.
The current study found significant differences in certain AAs, ACs, BAs, GPLs, and the Gln/Glu ratio between the MUO and MHO groups. So, we suggested in the 4. Discussion of a manuscript on the potential clinical implications of these metabolites on metabolic syndrome. Also, your comment is very important. Therefore, we have added more detailed text about the ‘5. Conclusions’ in the manuscript.
As per your suggestion, we have added text in the '5. Conclusions' section as follows:
(Line 502–511 in the revised manuscript):
5. Conclusions
The current study found significant differences in certain AAs, ACs, BAs, GPLs, and the Gln/Glu ratio between the MUO and MHO groups. In addition, the study observed that His, Lys, PCaaC34:1, and several clinical factors in adolescents of the MUO group were reverse correlated with the results in adolescents of the MHO group. The TyG index also was related to MUO in adolescents compared with HOMA-IR. In particular, it was associated with C2, Ala, Glu, Gln/Glu ratio, kynurenine, Met-SO, C34:1, HOMA-IR and TyG index in MUO adolescents. Therefore, these biomarkers found in this study have the potential to reflect the clinical outcomes of MUO in adolescents. These biomarkers will lead to a better understanding of the specificity and properties of MetS in obese adolescents.
Overall, the study investigates an important and timely topic and the results provide valuable insights into the differences between metabolically healthy and unhealthy obesity in adolescents. However, some additional context and clarification in the manuscript would greatly improve its readability and overall impact.
Author's response: Thank you for your kind and careful comments.
We have edited the manuscript and have corrected the English grammar. Also we added some text to acknowledgment in the ‘Ackowledgments’ section as follows:
(Line 537–538 in the revised manuscript):
Acknowledgments
We would like to thank Editage (www.editage.co.kr) for English language editing.
Thank you for considering my comments.
Submission Date
26 December 2022
Date of this review
08 Jan 2023 03:57:47
Thank you again for your profound advice, and we hope that our revision meets your requirements for publication as a final revision.
Sincerely yours,
Hye-Ja Lee, Ph.D.
Division of Endocrine and Kidney Disease Research, Department of Chronic Disease Convergence Research, Korea National Institute of Health, Korea Disease Control and Prevention Agency, 187 Osongsaengmyeong 2-ro, Osong-eup, Cheongju-si, Chungcheongbuk-do, 28159, Republic of Korea
Fax: +82-43-719-8709, Tel: +82-43-719-8452, E-mail: hyejalee@yahoo.co.kr

Round 2
Reviewer 1 Report
The study by Sarang Jeong et al. that presents a cross-sectional study aimed to identify potential metabolic biomarkers related to metabolically unhealthy obesity (MUO) in Korean obese adolescents was relatively well re-organized and the data presented are more clear.